# Research on a Hybrid Intelligent Method for Natural Gas Energy Metering

**DOI:** 10.3390/s23146528

**Published:** 2023-07-19

**Authors:** Jingya Dong, Bin Song, Fei He, Yingying Xu, Qiang Wang, Wanjun Li, Peng Zhang

**Affiliations:** 1Natural Gas Research Institute, PetroChina & Southwest Oil and Gas Field Company, Chengdu 610213, China; djy_xyz@sina.com (J.D.); songb@petrochina.com.cn (B.S.); he_fei@petrochina.com.cn (F.H.); xu_yy@petrochina.com.cn (Y.X.); wang.qiang@petrochina.com.cn (Q.W.); liwj@petrochina.com.cn (W.L.); 2School of Mechatronic Engineering, Southwest Petroleum University, Chengdu 610500, China; 3Key Laboratory of Natural Gas Quality Control & Energy Metering Measurement for State Market Regulation, Chengdu 610095, China

**Keywords:** natural gas, energy metering, artificial neural network, ultrasonic, accuracy

## Abstract

In this paper, a Comprehensive Diagram Method (CDM) for a Multi-Layer Perceptron Neuron Network (MLPNN) is proposed to realize natural gas energy metering using temperature, pressure, and the speed of sound from an ultrasonic flowmeter. Training and testing of the MLPNN model were performed on the basis of 1003 real data points describing the compression factors (Z-factors) and calorific values of the three main components of natural gas in Sichuan province, China. Moreover, 20 days of real tests were conducted to verify the measurements’ accuracy and the adaptability of the new intelligent method. Based on the values of the Mean Relative Errors and the Root Mean Square errors for the learning and test errors calculated on the basis of the actual data, the best-quality MLP 3-5-1 network for the metering of Z-factors and the new CDM methods for the metering of calorific values were experimentally selected. The Bayesian regularized MLPNN (BR-MLPNN) 3-5-1 network showed that the Z-factors of natural gas have a maximum relative error of −0.44%, and the new CDM method revealed calorific values with a maximum relative error of 1.90%. In addition, three local tests revealed that the maximum relative error of the daily cumulative amount of natural gas energy was 2.39%.

## 1. Introduction

Accuracy of measurement is essential to maintain natural gas trade fairness and to maintain pipeline network safety management, as well as correct billing for consumption [1]. In modern transmission networks, natural gases from different origins are transported and distributed, leading to diverse calorific values and physical property parameters among natural gases. In addition, in order to support the development of renewable energy sources, the Chinese government strongly promotes the efficient use of hydrogen resources [2,3]. It is expected that hydrogen mixing will further aggravate calorific value differences between natural gases. Consequently, the billing method of natural gas will need to be based on gas energy instead of volumetric measurement results, due to the wide variety of chemical characteristics in the networks.

Direct energy metering devices for natural gas are still not available on the market. The process for measuring natural gas energy requires both volume flow measurement using flowmeters, such as the orifice flowmeter, Venturi flowmeter, ultrasonic flowmeter, etc., and determination of the calorific value, which is calculated using the mole fractions of different components detected using a gas chromatograph. Therefore, the accuracy of the energy measurement of natural gas is a function not only of the volumetric measurement device used, but also of gas chromatography and of the measurement process. However, an online gas chromatograph is expensive, both in cost and in maintenance, to equip to every metering station [4,5].

The traditional calculation methods for the physical properties of natural gas, such as calorific value, compression factors (known as Z-factors), and energy measurement, are inseparable from the detection of natural gas components, with the exception of gas chromatography; another approach to determine the characteristics of gas components is the use of Raman spectroscopy. This can be used to assess natural gas thermodynamic parameters under different laboratorial environments [6,7,8,9]. Raman spectroscopy techniques have been validated for use in natural gas reservoirs for in situ composition tests for shale collection and for the detection of traces of water at sufficient levels in gas pipelines, as well as the detection of several contaminants [10,11]. However, as the main weaknesses of long fiber optic applications in Raman spectrometry techniques are high noise and strong signal attenuation, the applicability of Raman spectrometry in the field is limited. 

The choice of an appropriate method depends on the purpose and the complexity of the analyzed problem. In recent years, overcoming the aforementioned challenges of conventional techniques has motivated many scholars to utilize data-driven techniques, such as artificial intelligence (AI)-based methods, for process control, process optimization, and prediction. Some of these intelligent methods, so-called “black box” models, are data-driven methods which use sets of input and output data obtained from past data, data obtained during the measurement process, or data based on observed feature values (in regression, classification, and time series analysis). A typical AI-based technique is represented by artificial neural networks (ANNs). The prerequisites for using neural networks are sufficient historical data describing the problem and the existence of a true relationship or set of relationships between variables. Based on these historical data and mechanical relationships, the machine (computer) uses an algorithm combined with statistical examination techniques to build a model that can predict the output values without knowledge of all of the data, or with only a proportion of the related data.

ANNs have been used to support parameter prediction, e.g., natural gas load prediction [12], solar radiation potential [13], or thermal loads in regional energy systems [14,15]; process optimization, e.g., the parameter optimization of industrial gas turbines [16] or biodiesel-based engine performance [17]; and process control, e.g., fault detection [18] or steam turbine heating [19]. ANN models and their improved forms have gradually become more widely used in flowmeter verification, calibration, and measurement accuracy improvement. For example, B. Yazdanshenasshad and M.S. Safizadeh [20] built an ANN model to improve ultrasonic flowmeter calibration accuracy at two points. Xiong Yin et al. [21] built a hybrid AI model, which included a classic proportional integral differential (PID) controller and a genetic algorithm (GA) model, to optimize the verification process of a flowmeter; a radial-based artificial neural network was established [22] to improve the measurement accuracy of ultrasonic flowmeters and reduce the influence of flow field disturbances on measurement accuracy.

ANNs can also be used in the calculation of the thermodynamic state of fluids. Unlike the traditional natural gas thermodynamic parameter determination method, which requires knowing the components of the natural gas, an ANN-based approach can predict and measure natural gas components using some known thermodynamic parameters [23,24], based on data that were previously thought to be weakly correlated [25,26], or using incomplete data sets [27,28,29].

Recent studies have already proven that intelligent models have strong robustness and generalization. On the premise of knowing some components of natural gas or some other parameters (possibly from other sensors), an intelligent model can be used to solve for the desired natural gas energy metering parameters (such as the compression factor and calorific value of the tested natural gas) [30]. However, the measurement of natural gas energy has some limitations.

(1) In the literature, most of the research is focused on the calculation of natural gas physical parameters (compressibility, density, the Joule–Thomson coefficient, etc.), mainly based on some temperature and pressure data. Few papers report on the enhancement of flowmeters for natural gas energy metering.

(2) Most of the intelligent methods used in the existing literature are insufficiently advanced because they ignore the possible disadvantages of overly disaggregating data. Few published works have considered the influence of network structure on accuracy, and most of the improvements made are in the combining of intelligent models. By optimizing the structure of the model and combining it with an optimization algorithm, a much simpler and more efficient model can be established.

(3) In previous research papers, the calculation of actual accumulated natural gas energy at the site is not reported.

To solve these problems, we carried out the following:

(1) We established a natural gas energy measurement database based on the actual production data of the Southwest Oil & Gas Field Company, through which a novel natural gas energy metering model could be proposed.

(2) We used local pressure transmitter data, temperature transmitter data, and speed of sound data to establish and improve the intelligent model. We adjusted the optimal structure to avoid overfitting.

(3) To verify the adaptability of the developed model of the ultrasonic flowmeter, the natural gas energy transient and accumulation in 20 days were calculated, and the results were compared with field measurements.

## 2. Materials and Methods

### 2.1. Energy Measurement Principle

Traditional natural gas online energy metering includes volume measurement, determined using a volumetric flowmeter, and calorific value assignment achieved using a chromatograph. The natural gas energy can be determined by multiplying the volume flow and the calorific value using the following equation:(1)$$E_{b}=Q_{b}\times H_{b}$$
where
(2)$$Q_{b}=K_{1}uA$$
(3)$$H_{b}=K_{1}\sum_{i=1}^{n}x_{i}H_{i}$$*E_b_*, *Q_b_*, and *H_b_* are the energy flow, volume flow, and calorific value at the base condition, which is a STANDARD CONDITION for the natural gas trade. *u* is the average flow velocity measured using the ultrasonic flowmeter; *A* is the pipeline cross-sectional area; *x_i_* is the gas mole fraction; *H_i_* is the calorific value of *i* components; and *K*_1_ is a convert coefficient, determined using the following equation:(4)$$K_{1}=\left(\frac{T_{b}}{T_{f}}\right)\cdot \left(\frac{p_{f}}{p_{b}}\right)\cdot \frac{1}{Z_{f}}$$
where *T* is temperature, *p* is pressure, and *Z* is the compression factor. Subscript *b* represents the base condition; hence, *T_b_* = 20 °C, *p_b_* = 101,325 Pa. Subscript *f* represents the WORKING CONDITION, which varies between different flow fields. *Z_f_* is compressibility under working conditions, which is a parameter determined by ISO20765 and GB 17747, and relies on natural gas component analysis.

From Equations (1)–(4), it can be seen that the energy measurement of natural gas is inseparable from the traditional ultrasonic flowmeter and gas composition analysis instruments. However, the purpose of this paper was to add an algorithm on the basis of the original ultrasonic flowmeter, so that the ultrasonic flowmeter can complete the calorific value calculation without relying on chromatographic analysis.

### 2.2. ANN Approach

The application of an ANN model is a popular tool for the recognition, classification, or forecasting of selected process parameters. The advantage of neural networks over a mechanistic process model is that no prior knowledge about functional relationships between the driving input variables and the target variable is needed. As a result, neural networks were particularly suitable for our study, as we wanted to build a model to measure the natural gas energy without knowing the natural gas components. There are many types of ANN model, differing in structure and operating principles. In this study, a feedforward ANN multilayer perceptron (MLP) was used.

The main purpose of this study was to achieve natural gas energy measurement using an ultrasonic flowmeter only; the proposed idea for achieving this is shown in Figure 1.

Subplot (a) of Figure 1 shows a typical natural gas ultrasonic flow metering process. According to the measurement principle, the gas ultrasonic flowmeter is used to measure the gas flow velocity by measuring the time difference of multi-channel ultrasonic signal propagation. Therefore, the speed of sound (SOS) of an ultrasonic flowmeter is an inevitable parameter to be measured when using a gas ultrasonic flowmeter to measure gas volumetric flow. Although no use is currently defined for sound speed, an ultrasonic flowmeter measures sound speed as well as flow speed in the current measurement system. Both values are recorded as references for volumetric flow measurement accounting. In one recent study, Farzaneh-Gord et al. [26] developed an ANN method to calculate the natural gas compressibility factor and speed of sound. The results of their work showed that more than 95% of the data had good consistency. Therefore, it can be determined from the propagation law of SOS that ultrasonic sound velocity is different for different components, and can be used for characterizing gas composition.

Natural gas calorific value at base condition was determined using the proposed MLPNN model in this study, shown in subplot (b) of Figure 1. The number of neurons in the input layer was three, which was equal to the number of independent variables: temperature (*T*), pressure (*p*), and speed of sound (SOS); meanwhile, the number of neurons in the output layer was equal to the number of dependent variables.

Training of the MLP model was conducted on a set of 1003 cases, which were randomly separated into two categories for developing the models:

(a) Training set: 80% of the data were selected for model preparation, optimum discovery, and model prediction.

(b) Test set: this subgroup contained 20% of the cases of the whole dataset, utilized to assess the estimation ability of the model.

The training set was used to modify the neuron weights, while the test set was used for ongoing monitoring of the learning process. In the preliminary study, the different activation functions (logical function, linear function, exponential function, and hyperbolic tangent function) of neurons in the hidden layer and output layer were tested. The whole process of model calculation is shown in Figure 2.

As can be seen from Figure 2, the data were first randomly divided into a training set and a test set. ANN models were established and trained with the data of the training set until the training results met the stop requirements, and then, the test set was used for overfitting analysis to finally output the optimal model. Among these optimal models, a CDM model was established in this paper to combine the advantages of different ANN models to improve the accuracy of calculation. Details of the parameters of the above neural network are shown in Table 1.

As shown in Table 1, the obtained values in the hidden and output neurons were transmitted through an activation function. The activation functions are represented in Equations (5) and (6):

Tansig:(5)$$f\left(y\right)=\frac{-\exp\left(-y\right)+\exp\left(y\right)}{\exp\left(-y\right)+\exp\left(y\right)}$$

Purelin:(6)$$f\left(y\right)=y$$
where *y* represents the independent variables in the activation functions. Therefore, it can be seen that the model structures established in this paper were of the 3-*y_j_*-1 type. Input variables included temperature, pressure, and sound velocity; the hidden layer number was *y_j_*; and the output layer number was 1.

### 2.3. Working Steps

For the purpose of energy metering natural gas using an ultrasonic flowmeter, a novel accurate method is proposed to calculate the natural gas Z-factor and the calorific value with the knowledge of temperature, pressure, and ultrasonic wave velocity. The procedure of natural gas energy metering proposed in this study is shown in Figure 3.

As Figure 3 shows, the working steps were as follows:

Step 1: The first stage’s primary purpose was to establish a database, including working temperature (*T_f_*), working pressure (*p_f_*), mole fraction of composition (*x_i_*), volumetric flow (*Q_b_*), energy flow (*E_b_*), speed of sound (SOS), and other parameters calculated from the data collected by the local sensors, such as the compression factor (Z-factor), calorific value, etc.

Step 2: The main purpose of this step was to establish a new intelligent model to obtain the calorific value of the natural gas determination without a gas chromatograph. ANN models were built to calculate Z-factors and calorific values of natural gas with input data temperature (*T_f_*), pressure (*p_f_*), and speed of sound (SOS). Consequently, the artificial neural network (ANN) and its sub-methods (multilayer perceptron (MLP)) were trained and structured to calculate the natural gas energy.

Step 3: This step involved the evaluation of the forecast results. The overall instantaneous and accumulated energy were analyzed, considering the model’s measurement accuracy and pattern adaptability.

## 3. Validation of the New Model

### 3.1. Data Collection

The essential elements used in metering natural gas energy can be divided into climate variables, network working variables, and flowmeter working variables. The original dataset used in this paper was collected from the PetroChina Southwest Oil & Gas Field Company, who implement gas field development, gas marketing, pipeline real-time remote monitoring operations, and the allocation of gas field gathering and transportation networks in the Sichuan–Chongqing region.

In this study, a database of working pressure, temperature, and speed of sound for various natural gas compositions was created. The original dataset, used to establish and verify the model, included (i) shale gas from southern Sichuan; (ii) tight gas from the Jurassic Shaximiao Formation; and (iii) conventional natural gas from the Dengying Formation of the Sinian series gas field in central Sichuan. All data were repeatedly verified during the production run. The working conditions during the experiments are listed in Table 2, and the gas components are given in Table 3.

In this study, a total of 1003 samples were collected. The main research hypothesis considered that the external environmental conditions (i.e., humidity, atmospheric pressure, atmospheric temperature) of the measuring points were the average conditions of the study period. Hence, the calorific value of natural gas at a given measurement point could be predicted using the new neural network model in fixed external environmental conditions.

### 3.2. Validation of Z-Factor Results

According to Equations (2)–(4), Z-factors were required as key parameters for converting the natural gas energy in working conditions to the base condition. Usually, a large-scale measurement station equips an online chromatograph to achieve sampling, testing, and gas composition analysis. After this, equations of state, e.g., AGA8-92DC [31] recommended by ISO 20765-1, GERG-2008 [32] recommended by ISO 20765-2, or thermodynamic equations under some special conditions, are applied to determine the natural gas Z-factors in a central flow computer.

In this section, we employed four improved MLPNNs, including the Gradient Descent (GD) algorithm [33], Conjugate Gradient Descent (CGD) algorithm [34], Levenberg–Marquardt (LM) algorithm [34,35], and Bayesian Regularization (BR) algorithm [36,37], to calculate the Z-factors, with the input data including only the temperature, pressure, and speed of sound collected from the ultrasonic flowmeter. Figure 4 reveals the influence of the number of hidden neurons (Mean Relative Errors (MRE%s) and Root Mean Square Errors (RMSEs) for the training set and test set) used to predict natural gas compressibility.
(7)$$\mathrm{Mean}\;\mathrm{Relative}\;\mathrm{Error}\;(\mathrm{MRE})\;\mathrm{MRE}\%=\frac{100}{N}\sum_{i=1}^{n}\left|\frac{X_{\mathrm{MLP},i}-X_{\mathrm{real},i}}{X_{\mathrm{real},i}}\right|$$
(8)$$\mathrm{Root}\;\mathrm{Mean}\;\mathrm{Squared}\;\mathrm{Error}\;(\mathrm{RMSE})\;\mathrm{RMSE}=\sqrt{\frac{1}{N}\sum_{i=1}^{n}{\left(X_{\mathrm{MLP},i}-X_{\mathrm{real},i}\right)}^{2}}$$Here, *X_real,i_* is the real value of the natural gas, *X*_MLP,*i*_ is the estimated value, *n* is the number of testing times, and *N* is the total number of cases. In Figure 4, *X* represents the Z-factor. MRE% reflects the mean value of the errors between the estimated value and the real value in finite calculations. The smaller the value, the higher the accuracy of the calculation. Moreover, the RMSE reflects the degree of discretization between the calculated value and the true value. The smaller the value, the smaller the degree of discretization.

Subplot (a) of Figure 4 shows that the MRE and RMSE decreased with the number of neurons in a hidden layer. The smaller the MRE and RMSE, the smaller the error and dispersion of the calculated value, and the better the model calculation result. As a result, the order of the learning results was as follows:

BR algorithm ≈ LM algorithm > CGD algorithm > GD algorithm.

Subplot (b) of Figure 4 shows that the RMSEs of the GD algorithm, CGD algorithm, and BR algorithm were largely independent of the number of hidden layers, while the LM algorithm showed overfitting when the number of neurons in the hidden layers exceeded five. The order of the testing results was as follows:

BR algorithm ≈ CGD algorithm ≈ LM algorithm > GD algorithm (if number of neurons in the hidden layer ≤ 5).

BR algorithm ≈ CGD algorithm > GD algorithm > LM algorithm (if number of neurons in the hidden layer > 5).

Considering the lowest MSE%, the lowest RMSE, the minimum time required (the learning and testing time increased with the number of hidden layer neurons), and the most economical computing memory requirement (the learning and testing memory bank increased with the number of hidden layer neurons), the MLP 3-5-1 network model was chosen for further analysis. In the 3-5-1 MLP structure proposed in this paper, 3 represents the number of input parameters, 5 is the number of hidden layers, and 1 is the output parameter, which represents the Z-factor in this case.

A comparison of the estimated Z-factors and actual Z-factors using the MLP 3-5-1 model for 80% datasets in the network learning process, and for 20% datasets in the network learning process, is shown in Figure 5, using the four different algorithms: GD (a), CGD (b), LM (c), and BR (d). An error distribution plot is a method for demonstrating the distribution of errors around the zero-error line, in addition to showing whether models demonstrate error trends.

In order to analyze the accuracy of all the models more comprehensively, the performance of the developed models in terms of correlation coefficient R, degree of fitting R^2^, maximum/minimum relative error %, and maximum/minimum |*Z*_MLP_ − *Z_real_*| are listed in Table 4.
(9)$$\mathrm{Relative}\;\mathrm{Error}\;(\mathrm{RE})\;\mathrm{RE}\%=\frac{X_{\mathrm{MLP},i}-X_{real,i}}{X_{real,i}}\times 100$$

Correlation coefficient for each piece of input data (R):(10)$$R=\frac{N\left(\sum_{i=1}^{N}X_{real,i}\cdot X_{\mathrm{MLP},i}\right)-\left(\sum_{i=1}^{N}X_{real,i}\right)\left(\sum_{i=1}^{N}X_{\mathrm{MLP},i}\right)}{\sqrt{N\sum_{i=1}^{N}{\left(X_{\mathrm{MLP},i}\right)}^{2}-{\left(\sum_{i=1}^{n}X_{\mathrm{MLP},i}\right)}^{2}}\cdot \sqrt{N\sum_{i=1}^{n}{\left(X_{real,i}\right)}^{2}-{\left(\sum_{i=1}^{n}X_{real,i}\right)}^{2}}}$$
(11)$$\mathrm{Degree}\;\mathrm{of}\;\mathrm{fitting}\;(R^{2})\;R^{2}=1-\frac{\sum_{i=1}^{N}{\left(X_{real,i}\cdot X_{\mathrm{MLP},i}\right)}^{2}}{\sum_{i=1}^{N}{\left(X_{\mathrm{MLP},i}\cdot \bar{X}\right)}^{2}}$$
where *X_real,I_* is the real value of the natural gas, *X*_MLP,*I*_ is the estimated value, *n* is the number of testing times, and *N* is the total number of cases.

Table 4 shows that the Bayesian Regularization optimization algorithm was the best method, with the highest training/testing/total accuracy. In this study, it successfully predicted almost all the measurements with the training and testing sets, because it regarded weights as Gaussian distributions, eliminating the fixed weight value in a reverse correction. The R and R^2^ of the total set reached 0.9901 and 0.9801, respectively, and the maximum relative error of the total set was −0.44%.

### 3.3. Optimization of Calorific Value Calculation

The calorific value of natural gas is one of the most important parameters of energy measurement. Based on Equations (2)–(4), the calorific value under reference conditions should be measured based on accurate determination of the compression factor and calorific value of natural gas under working conditions. Similar to the determination of Z-factors, Figure 6 reveals the influence of numbers of hidden neurons (Mean Relative Errors (MRE%s) and Root Mean Square Errors (RMSEs) for the training set and test set) that were used to predict the natural gas calorific value. It is worth noting that when the MLPNN model was established, the output neuron was the calorific value of the working state of natural gas, and the measurement needed to use Z-factors to convert the working state to the base condition. Therefore, the MLP model should first be used to fit the compression factor under the working state, and then, the heat under the base state.

Subplot (a) of Figure 6 shows that the MRE and RMSE decreased with the number of neurons in the hidden layer, and the order of the learning results was as follows: BR algorithm ≈ LM algorithm > CGD algorithm > GD algorithm.

Subplot (b) of Figure 6 shows that the RMSEs of the GD algorithm, CGD algorithm, and BR algorithm were largely independent of the number of hidden layers, while the LM algorithm also showed overfitting when the number of neurons in the hidden layers exceeded eight. The order of the testing results was as follows:

LM algorithm ≈ BR algorithm ≈ CGD algorithm > GD algorithm (if number of neurons in the hidden layer ≤ 8).

BR algorithm ≈ CGD algorithm > GD algorithm > LM algorithm (if number of neurons in the hidden layer > 8).

In conclusion, we selected for further analysis the MLP 3-8-1 network model. In the 3-8-1 MLP structure proposed in this paper, 3 represents numbers of the input parameters, 8 is the number of hidden layers, and 1 is the output parameter, which was the calorific value in this case. The error distribution plot is shown in Figure 7.

Figure 7 shows that the distribution of other divisions demonstrated that the GD algorithm was not an appropriate model compared to the other models. When looking for an appropriate step size to update the parameters, the gradient direction of the current point was directly selected each time. Therefore, the minimum point obtained this time may not be the minimum value in the direction searched before, which led to slow or even non-convergence.

The Bayesian regularization optimization algorithm performed best in fitting on the whole. In contrast, the normalized test set values showed a relatively low fit of between 0.3 and 0.6 compared with the CGD algorithm, and the normalized test set values showed lower fitness of between 0.6 and 0.8 compared with the LM algorithm. The scattering of the data for the CGD algorithm was near the Y = X line for medium calorific values, and the LM algorithm and BR algorithm had less error scatter for the low and high values.

In order to combine the adaptability of these three algorithms, a diagram of the CDM (Comprehensive Diagram Method) is displayed in Figure 8. Multiple linear regression was established and utilized to determine the optimal coefficients for the CDM model. Therefore, to estimate the calorific value of natural gas, the CDM model was proposed as follows:(12)$$\mathrm{CDM}=a_{1}D_{CGD}+a_{2}D_{LM}+a_{3}D_{BR}$$
where *a*_1_–*a*_3_ are the constants of the CDM model, and *D* is the result of the ANN model. Table 5 presents their values.

In Figure 9, good agreement between the points and the distribution of the prediction errors is shown around the zero-error line in the training and test sets. A good concentration of data at low, medium, and high data for both the training and test data is obvious. The plot shows the excellent performance of the model and its ability to assess the system’s calorific value. Moreover, the performance of the developed models, in terms of correlation coefficient (R), degree of fitting (R^2^), maximum/minimum relative errors, and maximum/minimum |*H*_MLP_ − *H_real_*|, is given in Table 6.

Comparing the values in the table, BR-MLPNN, as well as the proposed new CDM method, performed better than the other models. The proposed CDM method had the best performance among the intelligent models, with a maximum |*H*_MLP_ − *H_real_*| of less than 0.71 MJ/m^3^, and BR-MLPNN had sub-excellent performance, with a maximum |*H*_MLP_ − *H_real_*| of less than 0.91 MJ/m^3^. The R and R^2^ of the new CDM method indicated that the prediction set reached 0.9710 and 0.9898, and the maximum RE% was 1.9%.

## 4. Field Application

### 4.1. Case Description

The new method described in this paper was tested under real working conditions for 20 days. We recorded natural gas component data, instantaneous energy measurement data, and accumulated energy data for 20 days. Volume flow measurement data were recorded every minute, and component data were updated every 6–7 min.

Our main pipe network had a diameter of 813 mm. In order to ensure accuracy, it was divided into three parallel pathways in the metering station, each with DN300. Three ultrasonic flowmeters were equipped and shared with one chromatographic data source. The experiments lasted for 20 days, from 21 September 2022 to 10 October 2022. The average ambient temperature was 288.1 K and the ambient pressure was 95 kPa. The pipe diameter was DN300. The measuring station mainly received gases from the Dengying Formation of the Sinian series gas field located in central Sichuan. Natural gas components were collected from local chromatographic devices, as shown in Table 7. The estimated accuracies of the Z-factor, calorific value, and metering bias of natural gas instantaneous and cumulative energy were taken into consideration.

Figure 10 shows the experimental group and control group. Concise explanations of these techniques are given below:

(1) In the control group, a chromatographic instrument was used to record the natural gas mole fraction to calculate the natural gas calorific value and other thermodynamic properties. Ultrasonic flowmeters, temperature transducers, and pressure transducers were used to record the natural gas volumetric flow under working conditions. The natural gas volumetric flow and energy flow were determined by multiplying the volumetric flow by the calorific value, in the traditional way.

(2) In the experimental group, the metering system contained no chromatographic instruments. Unlike the control group, ultrasonics were used not only to measure natural gas volumetric flow, but also to record sound velocity. The local speed of sound, temperature, and pressure were then introduced into the proposed CDM model to calculate the natural gas calorific value. Consequently, the natural gas energy was estimated using the new method.

### 4.2. Test Results

The natural gas Z-factors were estimated using the 3-5-1 structure BR-MLPNN method, and the calorific values were estimated using the new proposed CDM method. The results calculated using the NN algorithm are shown in Figure 11.

It can be seen from Figure 11 that the proposed intelligent method was able to effectively estimate the natural gas compressibility and calorific value, with a measurement R^2^ of over 0.98. The cumulative amount of natural gas energy calculated was compared with the cumulative amount of natural gas energy during the 20 days, as shown in Figure 12.

Compared with the traditional method, the errors of the new intelligent method of each ultrasonic flowmeter were limited; the maximum relative errors were 2.39%, 2.16%, and −2.27%, respectively.

However, as the amount of data in intelligent models is important, more data covering wider ranges can result in a more efficient model and better predictions. For trade purposes, it is necessary to establish a complete traceability system to ensure the fairness of the exchange of goods. Consequently, it is recommended to experimentally measure more data points at a wider range of pressure, temperature, and gas composition values, and to use them for modeling.

## 5. Conclusions

In this research, 1003 data points were utilized to model the Z-factors and calorific values for the metering of natural gas energy using only an ultrasonic flowmeter. An extensive databank, which included the pressure, temperature, ultrasonic sound velocity, and mole fraction of natural gas, was gathered from different available sources for the Southwest Oil and Gas Field Company. Different intelligent approaches were employed in this study, namely the Gradient Descent algorithm, Conjugate Gradient Descent algorithm, Levenberg–Marquardt algorithm, and Bayesian Regularization algorithm. Then, a Comprehensive Diagram Method (CDM) was developed based on the three best intelligent models to calculate the natural gas calorific value. Theoretical analysis and field tests were carried out, and the outcomes of the study can be summarized as follows:

(1) In the process of predicting Z-factors, the BR-MLPNN 3-5-1 network showed a natural gas Z-factor maximum relative error of −0.44%.

(2) In the process of predicting the calorific values of natural gas, the new proposed CDM method achieved the best fitting effect, with an R of 0.9710. This method had a maximum error of less than 0.7 MJ/m^3^, and relative error of less than 1.9%.

(3) Field tests showed that the new intelligent model could perfectly predict the accumulation of natural gas energy. The results revealed that the maximum relative error of the daily cumulative amount of natural gas energy was 2.39%. A larger sample for data analysis to continuously improve the NN and a better traceability system will make this new concept applicable to the gas trade.

## Figures and Tables

**Figure 1 sensors-23-06528-f001:**
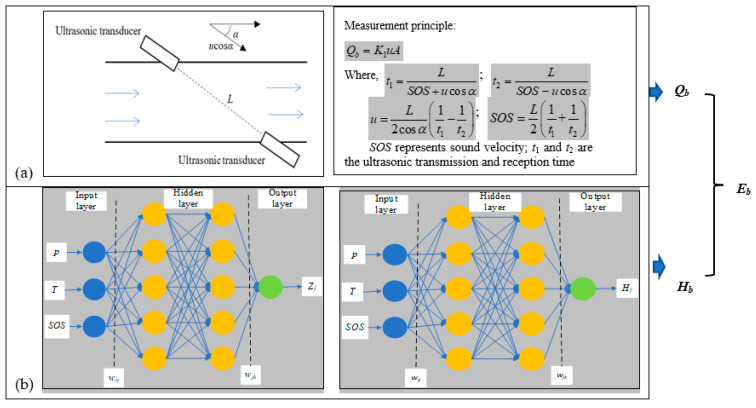
Design of the proposed new natural gas energy measurement method: (**a**) volume flow measurement principle of ultrasonic flowmeter; (**b**) artificial neural network method for calculating calorific value of natural gas.

**Figure 2 sensors-23-06528-f002:**
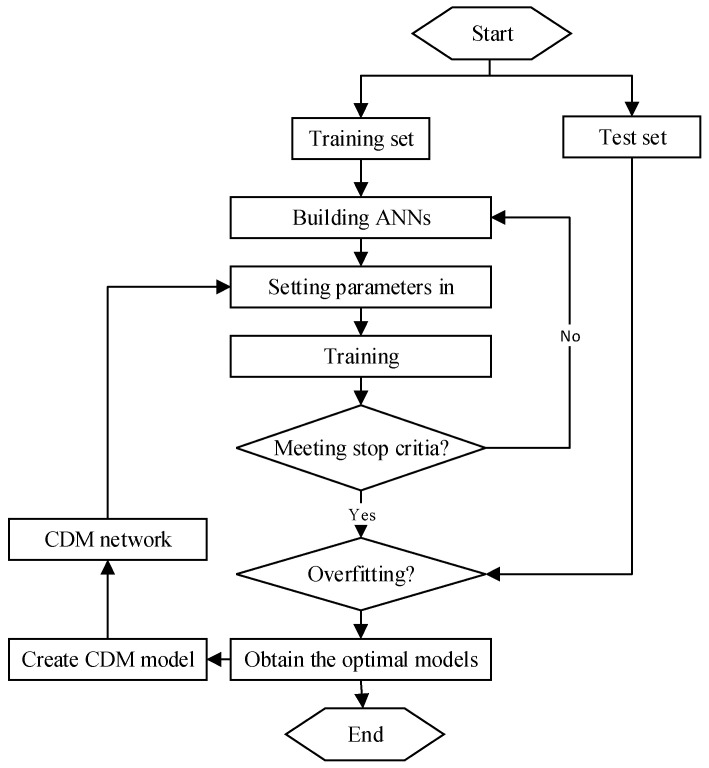
The whole process of model calculation.

**Figure 3 sensors-23-06528-f003:**
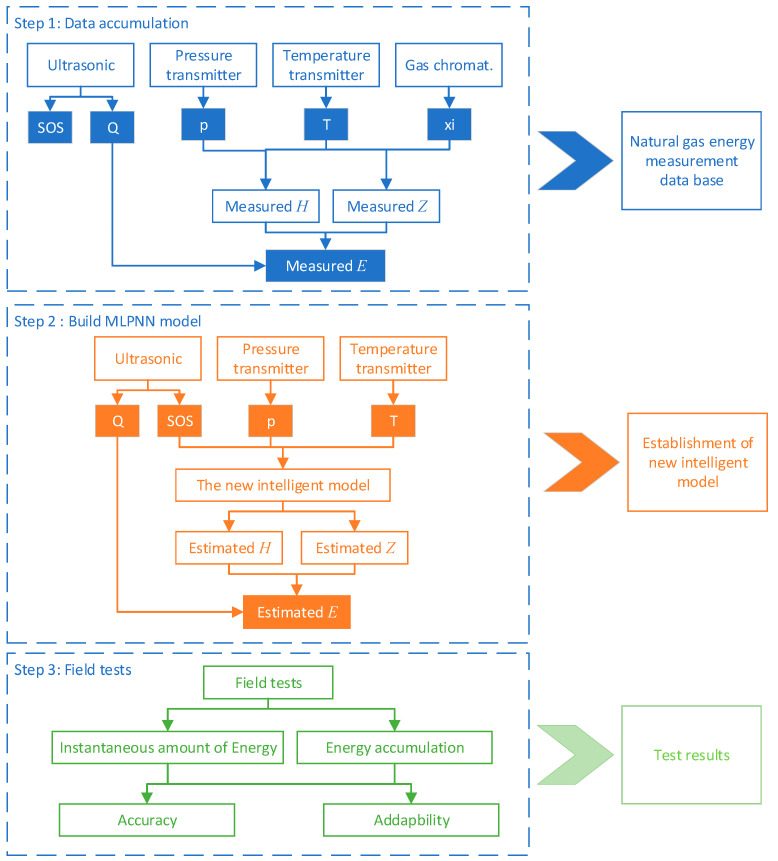
Procedure of natural gas energy metering proposed in this study.

**Figure 4 sensors-23-06528-f004:**
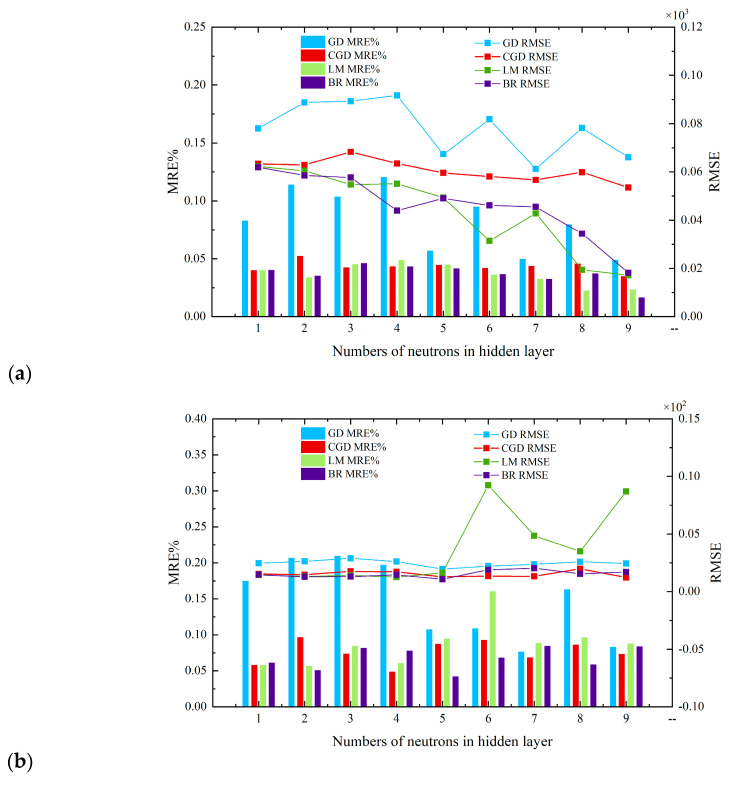
Summary of MRE and RMSE of Z-factors for different structures and for different algorithms of MLP network models for: (**a**) the training set; (**b**) the test set.

**Figure 5 sensors-23-06528-f005:**
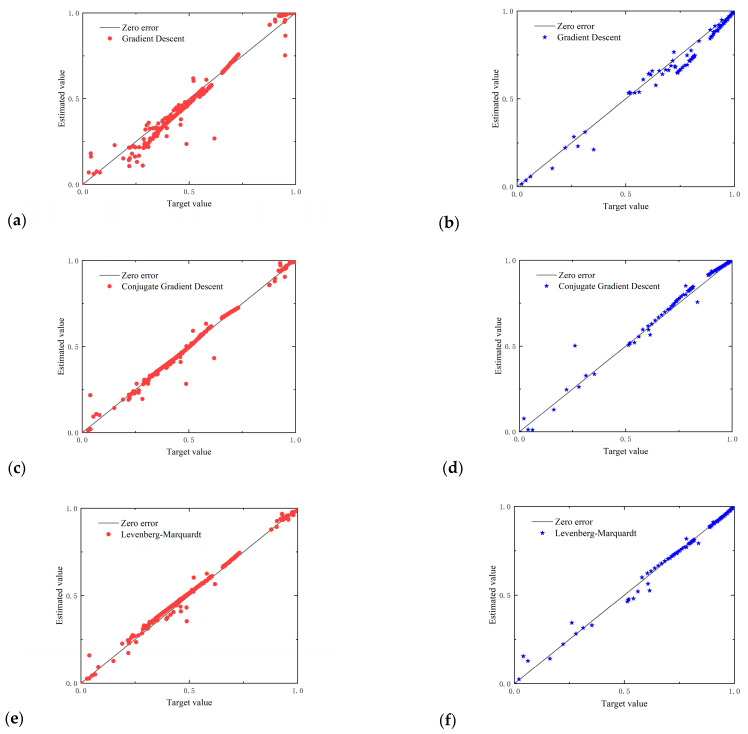
Comparison of the compressibility factor target data against MLPNN outputs for: (**a**,**c**,**e**,**g**) red dots—training set; (**b**,**d**,**f**,**h**) blue stars—test set.

**Figure 6 sensors-23-06528-f006:**
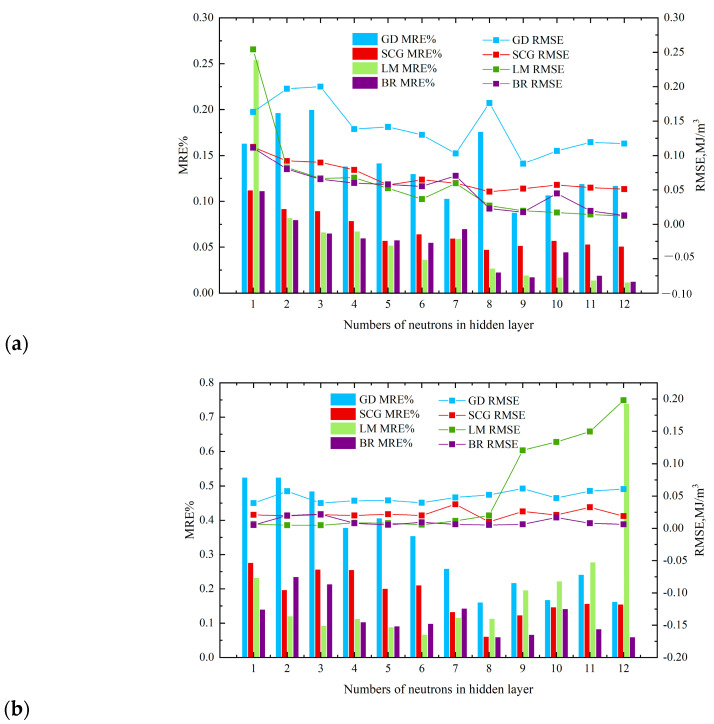
Summary of MRE and RMSE of calorific value for different structures and for different algorithms of MLP network models for: (**a**) the training set; (**b**) the test set.

**Figure 7 sensors-23-06528-f007:**
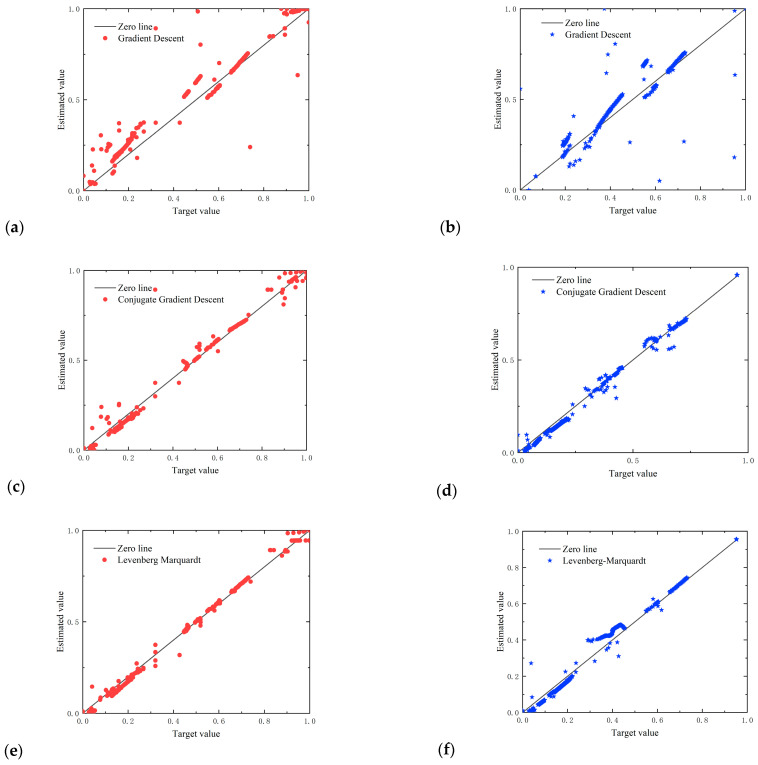
Comparison of the calorific value target data against the MLPNN outputs for: (**a**,**c**,**e**,**g**) red dots—training set; (**b**,**d**,**f**,**h**) blue stars—test set.

**Figure 8 sensors-23-06528-f008:**
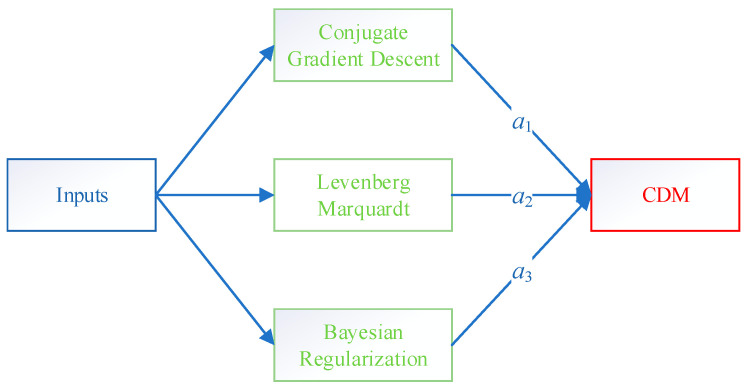
The CDM model proposed in this study.

**Figure 9 sensors-23-06528-f009:**
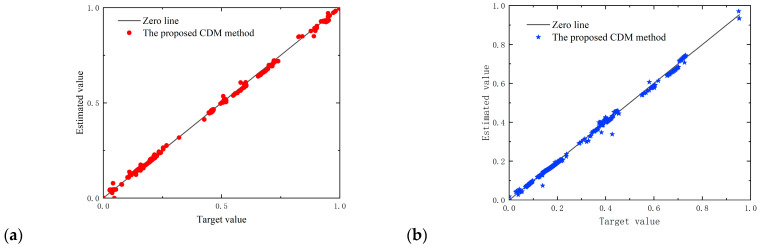
Comparison of the calorific value target data against the proposed CDM outputs for: (**a**) red dots—training data; (**b**) blue stars—test data.

**Figure 10 sensors-23-06528-f010:**
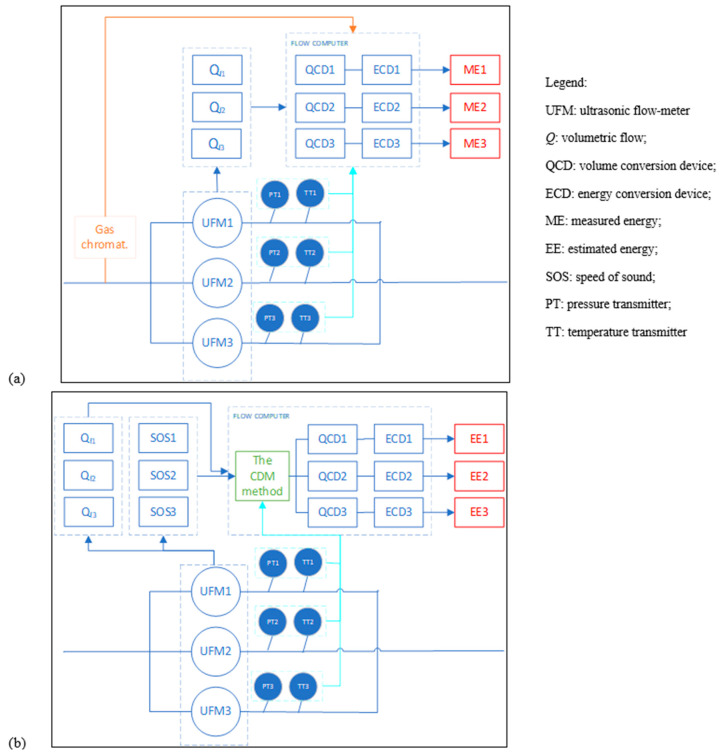
Schematic diagram of the test group and control group in the field trial: (**a**) the control group; (**b**) the test group.

**Figure 11 sensors-23-06528-f011:**
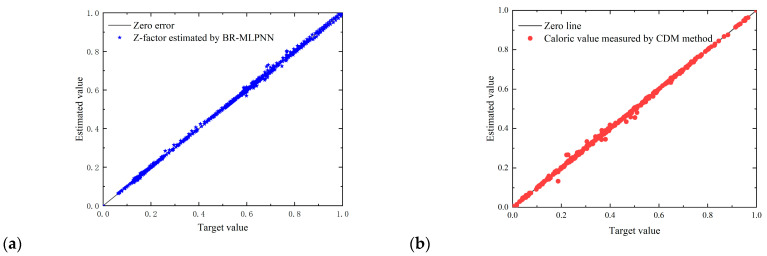
Comparison of normalized error: (**a**) Z-factors; (**b**) calorific values.

**Figure 12 sensors-23-06528-f012:**
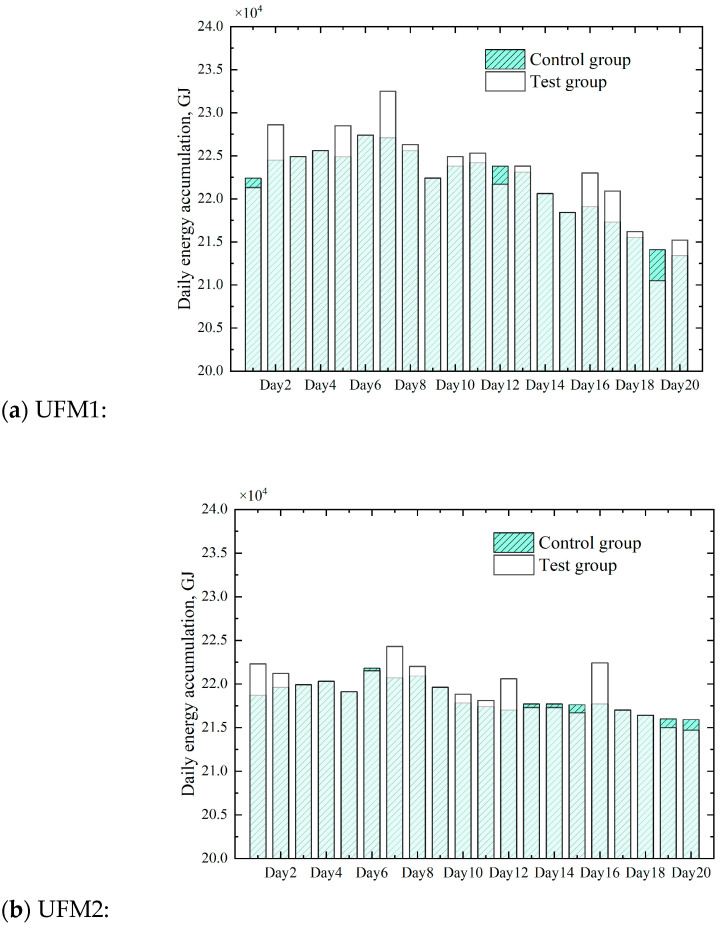
Comparison of results of natural gas energy accumulation of three ultrasonic flowmeters during the test time.

**Table 1 sensors-23-06528-t001:** A detailed report of the proposed MLPNN models.

Parameter	Compressibility	Calorific Value
Number of input layer neurons	3	3
Number of hidden layers	1~9	1~12
Number of output layers	1	1
Activation function of hidden layers	Tansig	Tansig
Activation function of output layers	Purelin	Purelin
Training data percent	80%	80%
Test data percent	20%	20%
Epoch	2000	2000

**Table 2 sensors-23-06528-t002:** Ultrasonic flowmeter working conditions.

Ultrasonic Working Conditions	
Atmospheric pressure, kPa	94.3~95.0 (average: 94.8)
Ambient Temperature, K	281.25~289.33 (average: 284.3)
Ambient Humidity, %RH	64.5
Diameter, mm	DN 300
Working gauge pressure, kPa	2762.2~5583.4
Working temperature, K	291.25~309.15
Speed of Sound, m/s	402.59~428.75

**Table 3 sensors-23-06528-t003:** Natural gas components.

Components	Max, %	Min, %	Average, %	SD
CH_4_, %	97.66	92.44	95.05	2.66
C_2_H_6_, %	1.28	0.05	0.665	1.31
C_3_H_8_, %	1.29	0.002	0.646	1.62
*i*C_4_H_10_, %	0.63	0	0.315	0.52
*n*C_4_H_10_, %	0.69	0	0.345	0.22
*i*C_5_H_12_, %	0.002	0	0.001	0.1
*n*C_5_H_12_, %	0.003	0	0.0015	0.1
*n*C_6_H_14_, %	0.32	0	0.16	0.22
N_2_, %	1.22	0.67	0.945	1.33
CO_2_, %	2.51	1.08	1.795	1.76
Calorific value, MJ/m^3^	37.65	35.79	36.24	3.43

**Table 4 sensors-23-06528-t004:** Comparison of error analysis of models in calculating Z-factors.

Algorithms		R	R^2^	Maximum RE%	Minimum RE%	Maximum |Z_MLP_ − Z*_real_*|	Minimum |Z_MLP_ − Z*_real_*|
GD	Training set	0.9696	0.9247	1.27%	0.0026%	0.0124	0.0000
	Test set	0.9562	0.7856	−1.93%	0.011%	0.0189	0.0001
	Total	0.9629	0.8407	−1.93%	0.011%	0.0189	0.0001
CGD	Training set	0.9850	0.9697	−0.80%	0	0.0078	0.0000
	Test set	0.9731	0.9670	1.18%	0.007%	0.0116	0.0001
	Total	0.9791	0.9681	1.18%	0	0.0116	0.0000
LM	Training set	0.9918	0.9763	0.73%	0	0.0072	0.0000
	Test set	0.9734	0.9681	0.98%	0.002%	0.0096	0.0000
	Total	0.9826	0.9723	0.73%	0	0.0072	0.0000
BR	Training set	0.9919	0.9871	0.33%	0	0.0032	0.0000
	Test set	0.9882	0.9774	−0.44%	0.004%	0.0043	0.0000
	Total	0.9901	0.9801	−0.44%	0	0.0043	0.0000

**Table 5 sensors-23-06528-t005:** The values of constants of CDM.

Range of Normalized Values	Constant	Value
0–0.3/0.6–1.0	*a* _1_	0.0029356
	*a* _2_	0.2127399
	*a* _3_	0.7843245
0.3–0.6	*a* _1_	0.7684462
	*a* _2_	0.0192079
	*a* _3_	0.2123459

**Table 6 sensors-23-06528-t006:** Comparison of error analysis of models in calculating the calorific value.

Algorithms		R	R^2^	Maximum RE%	Minimum RE%	Maximum |*H*_MLP_ − *H_real_*|, MJ/m^3^	Minimum |*H*_MLP_ − *H_real_*|, MJ/m^3^
GD	Training set	0.8493	0.7798	−4.75%	−0.08%	1.77	0.03
	Test set	0.8831	0.7213	3.07%	−0.06%	1.14	0.02
	Total	0.8662	0.7441	−4.75%	−0.08%	1.77	0.03
CGD	Training set	0.9440	0.9504	2.82%	0	1.05	0.00
	Test set	0.8804	0.8850	3.26%	0	1.21	0.00
	Total	0.9122	0.9221	3.26%	0	1.21	0.00
LM	Training set	0.9373	0.8947	1.78%	0	0.66	0.00
	Test set	0.8016	0.8433	3.22%	0.03%	1.20	0.01
	Total	0.8995	0.8765	4.22%	0.03%	1.57	0.01
BR	Training set	0.9676	0.9732	1.48%	0	0.55	0.00
	Test set	0.9255	0.9811	−2.44%	0.02%	0.91	0.01
	Total	0.9688	0.9862	−2.44%	0.02%	0.91	0.01
CDM method	Training set	0.9843	0.9902	1.02%	0	0.38	0.00
	Test set	0.9345	0.9830	1.90%	0	0.71	0.00
	Total	0.9710	0.9898	1.90%	0	0.71	0.00

**Table 7 sensors-23-06528-t007:** Natural gas composition, calorific value, relative density, and other parameters.

Name	Max	Min	Average	SD
CH_4_, %	97.8413	97.8344	97.8375	0.89
C_2_H_6_, %	0.1150	0.1120	0.1131	0.02
C_3_H_8_, %	0.0024	0.0023	0.0023	0.001
N_2_, %	0.7005	0.6748	0.6984	0.57
CO_2_, %	1.7828	1.4852	1.5532	0.82
Relative density	0.6885	0.6882	0.6884	0.06
Superior calorific value MJ/m^3^	36.33	36.24	36.29	0.19

## Data Availability

The data that were used in this study are confidential.

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
