# Peer review of "Research on a Hybrid Intelligent Method for Natural Gas Energy Metering"

_sensors, 2023, doi:10.3390/s23146528_

Round 1
Reviewer 1 Report
The paper is very interesting. In my opinion can be accepted in its present form
The methodology of the paper is very tight and rigorous, can help the current literature on the topic of the paper. Conclusions are consistent and the main question posed are well reported . The references are appropriate. Additional comments are that it’s a paper acceptable for publication in sensors
English is good
Author Response
Thank you very much for your recognition. For English language problems, we have made detailed revisions again, and the language changes in the whole paper are marked in blue. Thank you again.
Reviewer 2 Report
This paper presented a network that can be incorporated with an ultrasonic energy consumption monitoring system. The tested results are comparable with the chromatographic results, and would be of interest to readers in the relevant field. Therefore, I would recommend it for publication after following corrections/clarifications:
(1) The general writing needs to be improved: writings need tidy up, some sentences need to be reorganised for better understanding, long sentences can be separated, needs grammar check & spelling check...
(2) Are the input data pre-processed with feature extraction, or is it directly inputted with different training algorithms? Algorithms such as BR, LM and CGD are the training algorithms, they are not feature extraciton methods..?
(3) To build the network model, initial data were separated into training and testing, which I assume the testing means validation? If not, is there any reason to not have a validation dataset?
(4) Do provide a network topology/flow chart that is specific to the method described in the paper, including all the detailed processing steps. Figure 2 is too general.
(5) Does the ambient conditions affect the collected data? Is so, any compensation mechanisms?
See comments 1:
The general writing needs to be improved: writings need tidy up, some sentences need to be reorganised for better understanding, long sentences can be separated, needs grammar check & spelling check...
Author Response
Point 1: The general writing needs to be improved: writings need tidy up, some sentences need to be reorganised for better understanding, long sentences can be separated, needs grammar check & spelling check...
Answer: Thank you very much for your comments. We have checked the grammar and re-spelled the whole text and marked in blue. Thanks again.
Point 2: Are the input data pre-processed with feature extraction, or is it directly inputted with different training algorithms? Algorithms such as BR, LM and CGD are the training algorithms, they are not feature extraciton methods..?
Answer: Thank you for your opinions. For the input data, we delete the data that is obviously wrong, and then replace the missing data by linear interpolation. Algorithms such as BR, LM and CGD are training algorithms, not feature extraction algorithms.
Point 3: To build the network model, initial data were separated into training and testing, which I assume the testing means validation? If not, is there any reason to not have a validation dataset?
Answer: Thank you very much for your offer. For neural networks, there are generally two kinds of methods for processing initial data. First, as you mentioned, 70% of the total data is used as training data, 15% as test data and 15% as verification data. The other uses 80% of the total data as training data and 20% as test data,This method is also applied in the paper ‘Modeling the density of acid gases at extensive ranges of pressure and temperature conditions’ (Mohsen Riazi et al. 2021,207,109063). The purpose of this is: in the case of a small amount of data, a verification set is specially set to solve some hyperparameter or overfitting problems. In this paper, overfitting is considered separately, so in order to increase the training sample, 80% of the data is taken as the training set, and 20% is taken as the test set. Thank you again for your valuable advice.
Point 4: Do provide a network topology/flow chart that is specific to the method described in the paper, including all the detailed processing steps. Figure 2 is too general.
Answer: Thanks very much for your suggestion, we have redrawn Figure 2 to make the calculation process of the entire model more detailed and clearer. Thank you again for your suggestions.
Point 5: Does the ambient conditions affect the collected data? Is so, any compensation mechanisms?
Answer: Thank you very much. The compensation mechanism for ambient conditions is our future research direction. Because there are many environmental factors that affect the measurement accuracy of ultrasonic flowmeters, such as ambient temperature, electromagnetic interference, installation errors, etc., these factors will continue to be considered in future studies. Thank you again for your valuable comments.
Reviewer 3 Report
The paper on Research on a hybrid intelligent method for natural gas energy metering, by J. Dong et al., presents a detailed analysis of the measure of distributed natural gas by using a simplified measurement method and a tool based on ANN techniques. The study is rather complete using high quality data as reference to train and test the NN and, additionally, there is an application of the new tool in parallel with a reference measure on a real network conditions.
The paper presentation is correct in structure and content, with quality graphics and tables with enough information to support the discussions. The language has some flaws that disturbs the flow reading. Some comments are included, but the text has to be thoroughly check.
The paper deserves publication without any doubts. Still, before publications, the Authors should address a list of points to complete some information, or correct some parts. The Reviewer is convinced that the Authors can deal with this revision successfully to provide an updated version ready for publication.

The language has some flaws that disturbs the flow reading. Some comments are below, but the text has to be thoroughly check.
Many recommendations are included.
Author Response
Thank you very much for your suggestions.
-----------Abstract--------------
L13
Answer: Thank you very much for your advice. Firstly, Z-factors represent natural gas compression factors. According to the gas state equation, natural gas is compressible. Therefore, the conversion which using the Z-factors between operating conditions and base conditions should be carried out during measurement process to make all the trade fair. Secondly, the 'caloric value' in the article is replaced by 'calorific value'. All corrections are marked in red. Thank you again for your valuable advice.
--------Section -1-
L45
Answer: Thank you very much for your advice. We defined the Z-factors in both the abstract and the paper. Thanks again for your advice. We defined the Z-factors in both the abstract and the paper. Thanks again for your advice.
L 60
Answer: Thank you very much for your advice. We have verified and changed our expression again and marked it in red in the original text. Thank you again.
L63
Answer: Thank you very much for your valuable advice. We have revised it according to your suggestion and marked the original in red.
L133-130
Answer: We have re-write the whole part and marked it in red. Thank you for your valuable suggestions.
--------Section -2.1-
Answer: Thank you for your suggestions. Firstly, ‘base condition’ is a standard condition. In China, 20℃ and 101325Pa are base conditions. ‘Flow reference condition’ and ‘working condition’ represent the same meaning, which means conditions at a real flow. We have improved these statements one by one. Besides, we have re-write this part and added a Nomenclature to make the whole paper clearer. Thank you for your suggestions.
--------Section -2.2-
Answer: Thank you for your suggestions.
L159 have been removed, L163, L170 and L180 have been complete and marked in red.
We have defined A in section 2.1 and in Nomenclature part, t1 and t2 in Section 2.2.
L183 and L184 have been corrected and marked it in red.
We have re-drawn and re-write fig.2 part and marked it in red.
We have defined ‘Tansig’ and ‘Purelin’ and marked in red.
L210 have been deleted because we have moved the overfitting explanation in fig.2 part.
L216 have been deleted.
We have turned ‘operating condition’ into ‘working condition’.
L258 have been re-write and marked in red.
L260. The average condition and fixed condition represent the environmental condition, because environmental conditions can slightly reflect the measurement results.
L270. We have checked our ‘conditions’ expressions all over the paper to make the full text unified expression.
L277. We have added the references and marked in red.
L280. We have completed the sentence and marked in red.
L281. We have corrected the misrepresentation, and explained X. MRE and RMSE are briefly explained in the text.
L299. We have defined ‘order of learning’ in the text and marked in red.
L300 I'm terribly sorry. It's a clerical error on our part. We have corrected all the mistakes in the whole text.
L312 We have defined the model of this 3-yi-1 structure in Section 2.2, where we explain the model structure again and marked in red.
L316 We have corrected it. Thank you very much.
L323 We have removed this sentence, thank you very much for your suggestion.
L336 We have corrected this mistake.
L342-3 We have checked and corrected this mistake all the text, thank you for your suggestion.
----------Section 3.3------------
We have corrected it and marked in red.
L 374. We have corrected it, thank you very much.
L379 &Figure 8. We have corrected it, thank you very much.
L404 We have corrected it and marked in red, thank you very much. Hesi has been corrected, and written as HMLP.
L419&Fig.10. Our main pipe network has a diameter of 813 mm. During the measurement process, in order to ensure accuracy, it is usually divided into 3 parallel ways, and the diameter of each way is about 300mm (some are slightly greater than 300mm, and some are slightly less than 300mm, which needs to be determined according to the actual situation, but they are all DN300 pipelines). Although the chromatographic device can have multiple sampling points, according to the working principle of chromatography, it can only analyze the components of the most important sampling point in a period of time, so the three ultrasonic flowmeters are shared with one chromatographic data.
We have explained it in the text and marked in red. Thank you very much for your valuable suggestions.
L423 We have corrected it and marked in red, thank you very much.
L449 All data are calculated by NN algorithm directly and we have explained it in the text and marked in red. Thank you very much.
Fig.12 In the control group, natural gas energy was calculated by ultrasonic flowmeter and gas chromatographic instrument. While in the test group, natural gas energy was divided into the volume measurement part and the calorific value calculation part. The volume measurement part uses the on-site ultrasonic flowmeter data, and the calorific value and Z-factor are calculated by NN method. Due to the limited computing power of the PC, the cumulant calculation was not carried out using the cumulant per second, and the instantaneous amount is considered to be a uniform value for a period of time, resulting in the difference error distribution in the daily cumulant. Thank you very much for your worthy suggestions.
L490 We have changed this sentence and marked in red, thank you very much for your suggestions.
-------SYNTAX & LANGUAGE CHECK--------
Answer: We have corrected our spelling errors and marked in blue, thank you very much for your valuable comments.

Round 2
Reviewer 2 Report
The authors had addressed the comments carefully, and the overall quality of the manuscript has been improved. Therefore, I would recommend it for publication in its present form.
Reviewer 3 Report
The Reviewer acknowledges the thorough revision made by the Authors to provide an updated version of the paper.
The paper was revised with the list of points marked about language. A thorough reading and update of minor details before publishing will help to improve the draft.